# Promptriever: Instruction-Trained Retrievers Can Be Prompted Like Language Models

**Orion Weller** [*][ι]    **Benjamin Van Durme** [ι]    **Dawn Lawrie** [ι]

**Ashwin Paranjape** [α]    **Yuhao Zhang** [α]    **Jack Hessel** [α]

[ι] Johns Hopkins University    [α] Samaya AI

oweller@cs.jhu.edu

## Abstract

Instruction-tuned language models (LM) are able to respond to imperative commands, providing a more natural user interface compared to their base counterparts. In this work, we present Promptriever, the first *retrieval* model able to be prompted like an LM. To train Promptriever, we curate and release a new instance-level instruction training set from MS MARCO (Nguyen et al., 2016), spanning nearly 500k instances. Promptriever not only achieves strong performance on standard retrieval tasks, but also follows instructions. We observe: (1) large gains (reaching SoTA) on following detailed relevance instructions (+14.3 p-MRR / +3.1 nDCG on FollowIR), (2) significantly increased robustness to lexical choices/phrasing in the query+instruction (+12.9 Robustness@10 on InstructIR), and (3) the ability to perform hyperparameter search via prompting to reliably improve retrieval performance (+1.4 average increase on BEIR). Promptriever demonstrates that retrieval models can be controlled with prompts on a per-query basis, setting the stage for future work aligning LM prompting techniques with information retrieval. [1]

## 1 Introduction

Modern information retrieval (IR) models generally match queries to passages based on a single semantic similarity score. As a result, the user experience of search can be opaque, with users needing to find particular keywords/phrasings, apply various filters in advanced search settings, and iterate based on previous searches to find the "just right" query that returns the right passages.

In this work, we introduce *Promptriever*: a retrieval model that can instead be controlled via natural language prompts. For example, if a user is searching for James Cameron movies, but is only interested in movies prior to 2022 that are not co-directed, instead of applying a series of searches/hard filters, Promptriever can adjust its notion of relevance dynamically based on a natural language description: "Relevant documents are not co-directed, and are created before 2022" (see Figure 1 for a comparison).

Promptriever is a bi-encoder retriever; its backbone is a large language model (LM) such as LLaMA-2 7B (Touvron et al., 2023a;b).[2] Before IR training, language models can readily adapt their outputs based on natural language (also referred to as *instructions* or *prompts*). But after standard IR training, which typically focuses on optimizing a single query-passage "semantic similarity" score, instruction following capacity is not maintained (see §5). While some recent work has added instructions to retrieval model training sets (Su et al., 2022; Asai et al., 2022; Jiang et al., 2023; Muennighoff et al., 2024) these are templated "instructions", prepended to every query in the dataset at training and test time, rather than language defining relevance conditions on a per-instance basis. For example, when using the MS MARCO dataset, the standard "instruction" is *"Given a web search query, retrieve*

---

[*] Work performed during an internship at Samaya AI.

[1] Code and data are available at `https://github.com/orionw/promptriever`

[2] We experiment with multiple backbones in §5.1.

*relevant passages that answer the query"* which is prepended to every query (Muennighoff et al., 2024; Wang et al., 2023).

Promptriever, in contrast, is trained to maintain the per-instance instruction following capability of its backbone LM. To do so, we curate and release a synthetic dataset of ~500K query-passage relevance pairs augmented with instance-level instructions. The two key novelties in the training set are: (1) instructions defining per-query relevance in diverse, free-form natural language; and (2) instance-level *"instruction negatives."* Instruction negatives are cases where the (query, passage) pair is highly relevant if viewed in isolation, but, the addition of a carefully constructed instruction significantly decreases that relevance. For example, a generated instruction can request additional, fine-grained information not present in a topically relevant passage, as in Figure 2. By construction, to achieve low training loss on instruction negatives, models must adapt their notion of relevance per-query by the instruction, teaching them to be sensitive to fine-grained details.

Promptriever not only achieves strong retrieval scores in standard settings, but also follows instructions more effectively than prior models. Furthermore, we show the effectiveness of these methods by comparing to the same recipe with and without instructions. We show that instruction-tuning provides:

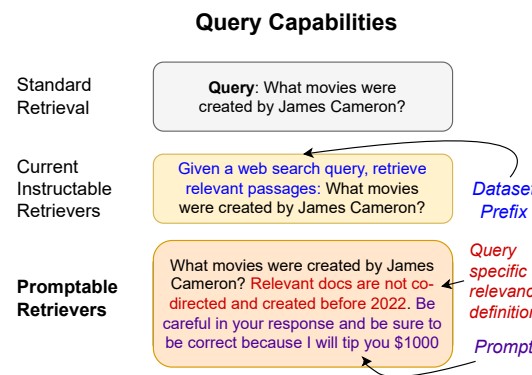

Figure 1: An illustration of the capabilities of retrieval models. Standard retrieval models find semantic similarity to the input query, typically matching using query keywords and phrases. Current instructable retrievers prepend a dataset prefix that generically describes the task, is used for all queries in the dataset, and is also generally used during training. We propose *promptable retrievers* which can handle complex instructions including detailed relevance definitions and zero-shot prompting techniques that act as a form of zero-shot hyperparameter optimization, just as prompting can be done with language models.

- State-of-the-art bi-encoder performance on instruction-following retrieval tasks (+14.3 p-MRR, +3.1 nDCG/MAP on FollowIR (Weller et al., 2024)) with comparable scores to SoTA cross-encoders.

- Improved robustness to query length/phrasing with a 44% decrease in variance across instructions on BEIR (Thakur et al., 2021) and +12.9 Robustness@10 on InstructIR (Oh et al., 2024).

- Reliable improvements in retrieval performance zero-shot solely by prompting (such as adding *"Think carefully when assigning relevance and I will give you a tip"*), enabling prompt engineering.

Our results demonstrate that with the right training data, modern bi-encoders *can* be instructed/prompted in free-form natural language, in a similar manner to LMs. We hope this alignment between the LM and IR communities will allow for further improvements to dense retrieval models.

## 2 PROMPTRIEVER DATA GENERATION

To train a bi-encoder that can retrieve based on instructions, we start with the popular IR training dataset, MS MARCO (Nguyen et al., 2016). We use the `tevatron-msmarco-aug` version which includes hard-negatives and was used to train RepLLaMA (Ma et al., 2023). This training set includes roughly 491k queries and provides a positive passage and 30 hard negatives for each. We augment this set with instructions in a two part process: (1) instruction generation from the initial queries and (2) instruction-negative mining. These processes are illustrated in Figure 2.

### 2.1 INSTRUCTION GENERATION

We start by generating an instruction[3] for each query in MS MARCO. Consider the example in Figure 2 where the initial query ("*Which type of volcano eruption has not been seen?*") is seeking

---

[3]See Appendix D for a definition of instructions.

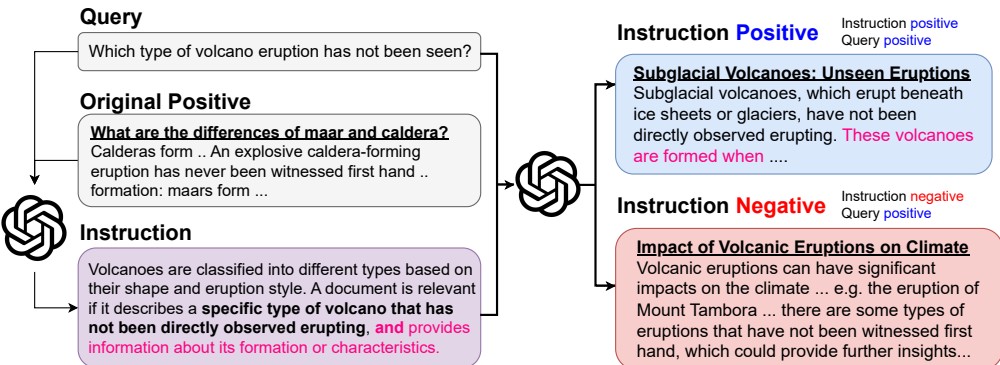

Figure 2: The data generation process to generate instruction-based retrieval data. We take the initial query and relevant passage and prompt an LM to generate an instruction that would match that query. Note that the instruction adds extra qualifications to the definition of relevance. We then ask an LM to generate an example relevant and non-relevant passage for that query *and* instruction. We see that the generated positive passage fulfills the extra requirement (in pink) but the generated instruction-negative does not. We generate multiple types of instructions (both in length and style).

just a *type of volcano.* We ask an LM to generate instructions that add additional requirements, explicitly exclude certain types of passages, or use ambiguity to make the initial query more specific. For example, in the volcano query, the generated instruction asks for both the volcano type *and* information about its formation (in pink). We use Llama-3-70B-Instruct (AI@Meta, 2024).[4]

**Instruction diversity**    To ensure a wide range of instructions, we ask Llama 3 to generate instructions of: 1) varying length formats (from short one-sentence instructions to two paragraph-sized[5] instructions), and 2) differing "styles", which can either be a persona of the person giving the query, negation of some aspect, or generic background information. The generated volcano instruction in Figure 2 has the generic background "style" and was requested to be two sentences long.

Overall, Llama 3 succeeded in following length+style specifications (Table 6). A qualitative exploration of examples is in the Appendix: by style in Table 8, and by length in Table 9.

**Maintaining original MS MARCO positive relevance**    We aim to generate instructions that maintain the positive relevance relation between (query, passage) pairs from MS MARCO. We provide both the query *and positive passage* to the LM when generating instructions, and request that the more specific instruction keep the passage relevant. To check for success in this regard: we use FollowIR-7B (Weller et al., 2024), a cross-encoder capable of making nuanced relevance judgments regarding (query, instruction, passage) instances. FollowIR-7B marked roughly 15% of the generated instructions as making the original positive passage no longer relevant. In these cases, we substitute the original positive document with one generated from the next stage.[6]

## 2.2    Instruction Negative Mining

After generating the instructions, it's possible to train models using the exact same data as RepLLaMA, except the queries have been augmented with instructions. However, our instructed-augmented queries have not changed any relevance relations in the corpus: with no additional modification, models could simply *ignore* our instructions entirely, and achieve the same performance.[7]

Thus, we develop a complementary data augmentation that encourages models to pay attention to the instruction. We term this augmentation *instruction negatives*:[8]  where the passage is query-positive

---

[4]See Appendix F for prompt details

[5]While we don't expect real users of IR systems to regularly type two-paragraph instructions, including these helps the model learn diversity of length and adapt to (potentially machine generated) diverse requests.

[6]We evaluated 20 of these instructions that passed the filter and found all positive passages were still relevant.

[7]As we show as an ablation in Table 4, in this setting, models indeed do ignore the instruction.

[8]This is similar to the intuition of Asai et al. (2022) but crucially the instruction-negatives are gathered at an instance level rather than at the dataset-level.

but instruction-negative, i.e., when the instruction is added it decreases the passage's relevance. To achieve low training loss, the model must learn to condition on both query *and* instruction.

Initial attempts to gather instruction negatives from the MS MARCO collection itself fell short: qualitatively, we found that the corpus does not contain enough positive passages per query to mine nuanced query-positives but instruction-negatives. Thus, we turn to generating passages with a LM.

We use `gpt-4o-2024-05-13` to generate the instruction negative passages, generating one query-positive/instruction-positive passage and three query-positive/instruction-negative passages per (query, instruction) pair. We over-generate candidates and then filter them post-hoc because initial testing revealed that (on average) only two out of three generated passages were correctly query-positive/instruction-negative.[9] We again use FollowIR-7B for the filter by: 1) checking that the generated instruction negatives are actually instruction-negative (and discarding it if not) and 2) checking that the generated instruction-positive was actually relevant (and discarding if not).

**Filtering validation**   We tasked 4 human annotators with the filtration task: for a given (query, instruction, generated passage) triplet, is the passage *relevant* or *not* to the query+instruction. For this task, average human-human agreement was 75% (N=32), whereas, the average human-model agreement was 84% (N=64). Thus, FollowIR-7B acts as a sufficiently high-quality filter.

## 3 EXPERIMENTAL SETTINGS

Our goal is to show the efficacy of instruction-training in IR compared to standard training. Thus, we primarily compare to RepLLaMA and use their data/recipe for apples-to-apples comparison.

### 3.1 TRAINING

We train Promptriever on the RepLLaMA MS MARCO data as well as the new instruction data generated by Llama 3 and GPT-4o. We use the same learning rate and other hyperparameter details as the original RepLLaMA for a fair comparison (see Appendix E for more details).[10] We use all valid instruction-negatives in training and sample the remainder of the hard-negatives from the dataset used to train RepLLaMA (keeping the same number of hard negatives per query).

### 3.2 EVALUATION DATASETS

We evaluate on in-domain (MS MARCO), out-of-domain (Thakur et al., 2021, BEIR), and instruction-following retrieval datasets, including InstructIR (Oh et al., 2024) and FollowIR (Weller et al., 2024). Note that these instruction-following datasets evaluate instructions on a *per-query* basis.

The metrics used are nDCG@10 for BEIR, TREC DL19 (Craswell et al., 2020) and DL20 (Soboroff, 2021), and MRR for MS MARCO Dev.

The instruction-following datasets use both standard and instruction-following metrics: nDCG@5 for the News21 portion of FollowIR, MAP@1000 for the Core17 and Robust04 portions, as well as using p-MRR for all portions (ranging from -100 to 100) which measures the sensitivity to instructions in the prompt (-100 means it follows the opposite of the instruction, 0 means no change, and 100 means perfect instruction following). InstructIR uses nDCG@10 as well as Robustness@10 which measures the minimum nDCG@10 score over 10 different prompts. Higher is better for all metrics.

We primarily compare with RepLLaMA to have an apples-to-apples comparison. However, we also show results for many other models, including MonoT5 (Nogueira et al., 2020), Instructor models (Su et al., 2022), the bi-encoder TART model trained from Contriever (Asai et al., 2022), E5 Mistral (Wang et al., 2023), Google Gecko (Lee et al., 2024), and BM25 (Robertson et al., 1995; Lù, 2024).

---

[9]We found that current LMs struggle with this nuance and that only the most capable LMs (e.g. GPT-4o but not GPT-4o-mini) were able to produce instruction negatives with high enough efficiency to be practical. As described in the previous section, we use the query-positive/instruction-positive passage as the backup positive passage if the original positive passage was not relevant to the generated instruction.

[10]Note that although this means that Promptriever has seen more data overall, we provide comparisons in Section 5 controlling for training data volume. Our findings do not change (i.e., that Promptriever's instruction following capacity is not simply a result of seeing more datapoints).

Table 1: Results for instruction following on the FollowIR and InstructIR datasets. Higher is better for all metrics; MAP@1000/NDCG@5/Robustness@10 range from 0-100; p-MRR ranges from -100 to 100. Despite using the same backbone model (Llama 2) and recipe, Promptriever significantly outperforms RepLLaMA with a +3.1 gain on the standard retrieval score (nDCG/MAP average) and a +14.3 point gain on p-MRR. **Promptriever outperforms all other dense retriever models and scores comparably to the best cross-encoder**, FollowIR-7B, despite not using attention between query and documents. Gecko scores use a proprietary API and were not reported for InstructIR. Bold results indicate the best for that architecture type (e.g. cross-encoder, bi-encoder) while asterisks are statistical similarity to the best (for bi-encoders only, as cross-encoders are not comparable §B).

| | Model | FollowIR | | | | | | | | InstructIR | |
| | | Robust04 | | News21 | | Core17 | | Average | | MS MARCO | |
| | | MAP | p-MRR | nDCG | p-MRR | MAP | p-MRR | Score | p-MRR | nDCG | Robust. |
|---|---|---|---|---|---|---|---|---|---|---|---|
| Cross Encoders | MonoT5-3B | **27.3** | +4.0 | 16.5 | +1.8 | 18.2 | +1.8 | 20.7 | +2.5 | - | - |
| | Mistral-7B-instruct | 23.2 | +12.6 | 27.2 | +4.8 | 19.7 | +13.0 | 23.4 | +10.1 | 63.1 | 35.3 |
| | FollowIR-7B | 24.8 | **+13.7** | **29.6** | **+6.3** | **20.0** | **+16.5** | **24.8** | **+12.2** | **81.3** | **71.5** |
| Bi Encoders | BM25 | 12.1 | -3.1 | 19.3 | -2.1 | 8.1 | -1.1 | 13.2 | -2.1 | 76.0 | 26.9 |
| | TART-Contriever | 14.3 | -9.0 | 21.8 | -3.0 | 13.3 | -3.0 | 16.5 | -5.0 | 84.8 | 47.5 |
| | Instructor XL | 19.7 | -8.1 | 26.1 | -0.9 | 16.8 | 0.7 | 20.9 | -2.8 | 48.6 | 21.5 |
| | E5-Mistral | 23.1 | -9.6 | 27.8 | -0.9 | 18.3 | +0.1 | 23.1 | -3.5 | 86.3 | 55.4 |
| | Google Gecko | 23.3 | -2.4 | *29.5 | *+3.9 | *23.2 | +5.4 | *25.3 | +2.3 | - | - |
| | RepLLaMA | 24.0 | -8.9 | 24.5 | -1.8 | 20.6 | +1.3 | 23.0 | -3.1 | 85.7 | 50.2 |
| | Promptriever | *28.3 | *+11.7 | *28.5 | *+6.4 | *21.6 | *+15.4 | *26.1 | *+11.2 | *92.1 | *63.1 |

# 4 RESULTS

Promptriever outperforms the original RepLLaMA in instruction following (§4.1) while maintaining strong standard retrieval performance (§4.2). We also demonstrate that Promptriever can be reliably zero-shot prompted, in the same manner as a language model (§4.3).

We further show that these are statistically significant, and compute a two-sided student t-test ($p < 0.05$) for nDCG and MAP metrics and Wilcoxon signed rank non-parametric test (due to a non-normal distribution) for p-MRR and Robustness@10 (also using $p < 0.05$)

## 4.1 INSTRUCTION FOLLOWING

Table 1 presents the results for the FollowIR and InstructIR datasets. Promptriever is the highest performing dense retriever, improving over RepLLaMA by +14.3 p-MRR (-3.1 $\rightarrow$ +11.2) and +3.1 in nDCG/MAP. For reference, we also include the results from three computationally intensive cross-encoder models. While cross-encoders (as expected) perform best due to their significant compute advantage, Promptriever achieves comparable scores as a much more efficient bi-encoder model. Our model's strong performance versus the RepLLaMA baseline illustrates that our instruction data is highly effective, leading to significant gains in instruction following and prompt robustness.[11]

## 4.2 IN-DOMAIN RETRIEVAL

We benchmark Promptriever on three standard retrieval tasks from in-domain data, e.g. MS MARCO based (Table 2). We see that Promptriever performs comparably to RepLLaMA on in-domain tasks despite additionally having stronger instruction following performance.

Table 2: MS MARCO (in-domain). We see that they are comparable, despite Promptriever being instruction-trained. Bold indicates best in the column, asterisks are statistical similarity to the best.

| Model | DL19 nDCG@10 | DL20 nDCG@10 | Dev MRR |
|---|---|---|---|
| RepLLaMA | **\*74.5** | *71.8 | **\*42.5** |
| Promptriever | *73.2 | **\*72.3** | 42.0 |

---

[11]It is likely that a cross-encoder trained on our generated data would outperform FollowIR-7B, however, we leave that for future work and focus only on dense retrievers in this work.

[12]Although standard NQ has a training set, the BEIR NQ version selects a different set of queries and documents and doesn't provide a comparable dev/train set, c.f. <removed for anonymity>.

Table 3: Out of domain performance on BEIR (nDCG@10). Promptriever performs similarly to RepLLaMA without any instructions (*None* column). However, when given a prompt we see that performance improves for Promptriever by +1.4 points whereas RepLLaMA and BM25 perform worse. **We thus see that Promptriever is promptable due to its instruction-training. We also see that it is possible to select the best prompt from 10 dev examples consistently** for Promptriever, as the difference between the *Selected Prompt* is almost always the same as the *Best Prompt* of the ten prompts evaluated. Note that not all BEIR datasets have training/dev sets and thus Selected Prompt is left blank for them. Best value in the row is bolded (see Appendix C for statistical tests).

| | Dataset | BM25 | | | RepLLaMA | | | Promptriever | | |
|---|---|---|---|---|---|---|---|---|---|---|
| | | None | Selected Prompt | Best Prompt | None | Selected Prompt | Best Prompt | None | Selected Prompt | Best Prompt |
| Dataset has dev/train set | DBPedia | 29.9 | 23.3 | 23.3 | 44.8 | 16.8 | 43.5 | 45.0 | **45.2** | **45.2** |
| | FEVER | 48.1 | 7.4 | 45.3 | 82.9 | **85.3** | **85.3** | 82.8 | 82.8 | 82.8 |
| | FiQA | 25.1 | 16.9 | 21.9 | 45.0 | 41.8 | 42.7 | 45.9 | **46.6** | **46.6** |
| | HotpotQA | 56.9 | 54.6 | 54.6 | 68.8 | 66.4 | 67.9 | 69.2 | **69.5** | **69.5** |
| | NFCorpus | 32.1 | 18.0 | 23.5 | 36.0 | 34.2 | 35.0 | 36.5 | **36.9** | **36.9** |
| | Quora | 80.4 | 75.3 | 75.3 | 86.0 | 83.1 | 85.5 | 86.5 | **88.0** | **88.0** |
| | SciFact | 68.7 | 65.7 | 65.7 | 75.3 | 75.0 | 75.7 | 75.0 | 75.9 | **76.3** |
| No dev/train set | Arguana | 36.6 | - | 36.4 | 48.6 | - | 49.0 | 51.8 | - | **56.7** |
| | Climate-FEVER | 13.6 | - | 13.9 | 29.3 | - | 30.8 | 27.6 | - | **32.1** |
| | NQ[12] | 28.5 | - | 25.4 | **63.0** | - | 62.2 | 61.9 | - | 62.6 |
| | SCIDOCS | 15.8 | - | 14.9 | 16.1 | - | 16.7 | 17.3 | - | **19.7** |
| | TREC-COVID | 62.3 | - | 35.9 | 83.9 | - | 82.7 | 83.9 | - | **84.6** |
| | Touche-2020 | 33.1 | - | 30.8 | 34.1 | - | **35.9** | 31.4 | - | 32.0 |
| | Average | 40.9 | - | 35.9 | 54.9 | - | 54.8 | 55.0 | - | **56.4** |

## 4.3 OUT-OF-DOMAIN RETRIEVAL

We use the BEIR benchmark to measure out-of-domain performance. We do so in two settings: (1) no prompts and (2) with prompts. For out of domain performance on BEIR (Table 3) **without prompts** (i.e. the *None* column) we also find comparable scores: Promptriever performs similarly to RepLLaMA (Promptriever averaging 55.0 vs. 54.9 from RepLLaMA).

However, we can also evaluate them with prompts. We use the common approach from the LM community, which seeks to improve LMs on out-of-domain data by including a textual prompt at test-time, even if the prompt is somewhat generic, e.g., "think step by step" or "I'll give you a tip"(Kojima et al., 2022; Wei et al., 2022b). We apply this approach to IR by exploring whether particular prompts reliably induce improved retrieval performance in Promptriever.

We use the following settings for testing prompts, following the standards in the LM community; typically one would evaluate prompts for an LM by first using a small validation set. We sample 10 queries from each of the validation (or train if there is no validation set) to use as the prompt tuning set. We also create 10 generic prompts[13] that could work across retrieval datasets.

However, not all of the BEIR datasets have train/dev data to sample validation examples from for selecting a prompt. We thus show results in two ways (Table 3): (1) when there is a dev set we select the best dev prompt as the test prompt (*Selected Prompt* column) and leave the score blank for datasets without a dev/train set; and (2) taking the best prompt of the ten (*Best Prompt* column).

We see in Table 3 that, for Promptriever, using the best prompt brings significant gains to BEIR[14] average performance (+1.4 nDCG@10; gains versus no prompt for 12/13 datasets and tied on the last). However, in contrast, prompts fail to bring any gains to the RepLLaMA or BM25 models with -0.1 and -5.0 nDCG deltas respectively. Thus we can see that prompting is effective for Promptriever but not for retrieval models using standard training.

---

[13]We include the generated prompts in Appendix A

[14]Individual prompts and their scores on each BEIR dataset are found in Table 11.

Table 4: Ablations for instruction following on the FollowIR and InstructIR datasets. **The instruction provides gains beyond the simple length of the instruction or its distribution**. Instruction-negatives and joint data bring even more gains. Best value is in the column is bolded while asterisks indicate statistical similarity to the best score.

| Model | FollowIR | | | | | | | | InstructIR | |
| | Robust04 | | News21 | | Core17 | | Average | | MS MARCO | |
| | MAP | p-MRR | nDCG | p-MRR | MAP | p-MRR | Score | p-MRR | nDCG | Robust. |
|---|---|---|---|---|---|---|---|---|---|---|
| RepLLaMA | 24.0 | -8.9 | *24.5 | -1.8 | *20.6 | +1.3 | 23.0 | -3.1 | 85.7 | 50.2 |
| Repeated Query | 24.6 | -9.1 | *25.3 | -2.6 | *21.1 | +2.4 | 23.6 | -3.1 | 85.4 | 49.2 |
| Generic Instruct | 25.5 | -7.2 | *26.2 | -1.7 | *21.6 | -0.0 | *24.4 | -3.0 | 63.1 | 32.4 |
| Swap Instruct | 25.2 | -1.9 | *27.3 | -0.2 | *21.1 | -0.6 | *24.6 | -0.9 | 48.6 | 27.0 |
| w/Instructions | *26.9 | +3.8 | *29.1 | *+5.3 | *20.7 | +8.0 | *25.6 | +5.7 | *91.9 | *63.3 |
| w/Instruction Negatives | *29.0 | *+9.7 | *27.8 | *+5.2 | *21.9 | +11.4 | *26.2 | +8.8 | *91.5 | *62.0 |
| Promptriever (Joint) | *28.3 | *+11.7 | *28.5 | *+6.4 | *21.6 | *+15.4 | *26.1 | *+11.2 | *92.1 | *63.1 |

But are these best prompt numbers close to what would be achieved in practice with a small dev set? The *Selected Prompt* column shows score from a practical setting. If we compare Promptriever's score for Selected Prompt vs Best Prompt, we see that there is very little difference between them. Applying few-shot selection with the dev set selects the best prompt in 6/7 cases, and in the 7th case (SciFact) it chooses a prompt that is still almost one full point better than the no prompt setting. In contrast, and as expected, BM25 is not "promptable" in any setting with performance dropping across the board; RepLLaMA's performance drops (sometimes dramatically) in six of seven cases.

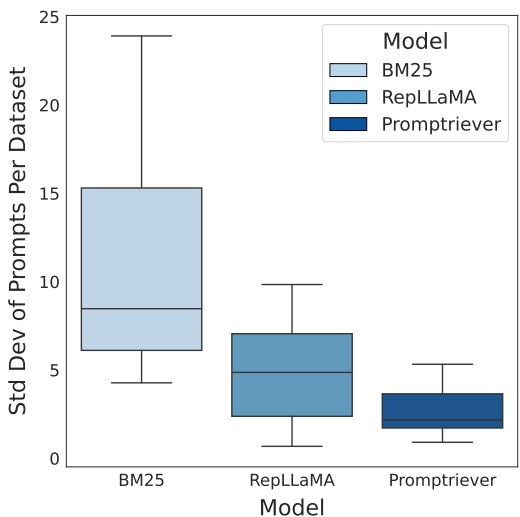

Figure 3: Standard dev. of NDCG@10 scores per dataset across all 10 prompts. We see that Promptriever is much more robust to the phrasing.

We also examine the sensitivity of all models to the prompts (Figure 3). We see that Promptriever's variance to prompts is significantly less than that of RepLLaMA (by 44%) and BM25 (by 77%) which has wide swings due to the effect of the keyword matching. This suggests that Promptriever is more robust to the input and is less sensitive to the exact keywords being used.

In summary, the standard practice of instruction-training with LMs can apply to instruction-training dense retrievers as well, but only if they, like Promptriever, are trained to be sensitive to such prompts. Furthermore, strong gains are indeed reliably possible to achieve via selecting a natural language prompt using a small held-out eval set, similar to how LMs are typically prompted.

## 5 ANALYSIS

We ablate several null hypotheses in an effort to better understand which parts of the Promptriever training recipe contribute most to performance gains. We train all of the ablated models on a dataset consisting of half standard MS MARCO data and half instruction-based MS MARCO data for compute and speed reasons (also matching the number of training instances in RepLLaMA). Results for all ablations are in Table 4, with the baseline being RepLLaMA, and each row of the table representing either a null hypothesis or a design decision leading to the final Promptriever models.

**Q1: Is it simply the length of the query/instruction that enables Promptriever's performance gains?** *Answer: No.* We train models with the query repeated to the length of the instruction

Table 5: Comparison of different backbone models on the same Promptriever recipe across MS MARCO datasets (DL19, DL20, and Dev), BEIR, InstructIR, and FollowIR. **We see that our augmented instruction dataset provides gains for many different base models, indicating the generality of our approach**. Hyperparameter tuning would likely improve these results for other backbone models. See Appendix G for a discussion on BERT-sized models.

| | Base Model | MS MARCO | | | BEIR | | FollowIR | | InstructIR | |
|---|---|---|---|---|---|---|---|---|---|---|
| | | DL19 | DL20 | Dev | nDCG | w/Prompt | Score | p-MRR | nDCG | Robust@10 |
| | RepLLaMA | 73.2 | 72.3 | 42.0 | 54.9 | 54.8 | 23.0 | -3.1 | 85.7 | 50.2 |
| Promptriever | LLaMA2 | 74.5 | 71.8 | 42.5 | 55.0 | 56.4 | 26.1 | +11.2 | 92.1 | 63.1 |
| | Mistral v1 | 72.9 | 73.4 | 42.3 | 54.4 | 55.7 | 25.7 | +11.8 | 90.3 | 58.8 |
| | Llama 3.1 | 73.5 | 72.9 | 43.2 | 55.1 | 56.5 | 25.0 | +11.3 | 85.5 | 41.6 |
| | Llama 3.1 Instruct | 72.4 | 73.6 | 42.7 | 55.5 | 57.2 | 26.0 | +9.8 | 89.9 | 57.8 |

(*Repeat Query*) and with *Generic Instructions*.[15] Increasing the length of queries results in a slight gain on standard retrieval performance, but little gain in p-MRR (retrieval sensitivity). This includes the *Repeat Query* (+0.6 nDCG/MAP) and *Generic Instruction* (+1.4 nDCG/MAP) baselines.

**Q2: Is it the lexical distribution of the instructions (in isolation) that enables these gains?** *Answer: Partially.* For this we train with the real generated instructions but randomly swap the instruction each query is paired with (*Swap Instructions* row). Compared to the length ablations, we see larger gains in the score (+1.6) and p-MRR (+2.1) as the model has learned the distribution, although not how to use them effectively.

**Q3: How much does training with instructions help?** *Answer: Significantly.* We ablate this by showing the results of training with just the instructions and no instruction-negatives (*w/Instructions*). We see a strong gain in p-MRR (+6.6) and a further gain in standard retrieval (+1) over Swap.

**Q4: How much does training with instruction-negatives help?** *Answer: Significantly.* Adding the instruction negatives on top of *w/Instructions* gives another large gain in p-MRR (+3.1 over *w/Instructions*) and a small boost in standard retrieval scores (+0.6 nDCG/MAP). This aligns with expectations: instruction negatives provide extra data for instruction sensitivity but not necessarily for standard retrieval metrics.

**Q5: Does training additionally on MS MARCO help beyond the Promptriever training set we curate?** *Answer: Yes. Promptriever (Joint)* our final model, combines all the standard MS MARCO and Instruction MS MARCO data which leads to another large jump in p-MRR (+2.4) as it is able to see more data (and instructions) in training, i.e. 2x as much.

**In summary,** each step in our final recipe (+w/ Instructions, +w/ Instruction Negatives, +MS MARCO Jointly) provides value independently, and that value is not due to simple factors like increasing the length of the query and/or surface lexical features of the instructions.

## 5.1 DOES THIS PROCESS WORK FOR OTHER MODELS?

The original RepLLaMA used Llama 2 as a backbone, and, to this point in our paper, Promptriever has also used Llama 2 as a backbone for fair comparison. We also adopt the same training hyperparameters as RepLLaMA. Nonetheless, we ablate different LM backbones to see if performance holds without any adjustments to the hyperparameters or training recipe. While further tuning the learning rate and other parameters would likely improve performance, we see in Table 5 that other backbones provide comparable performance, indicating the generality of our method.

---

[15]We add a generic retrieval instruction from one of 50 different generic retrieval task descriptions generated by `GPT-4o` and `Claude-3.5-Sonnet`. See a full list in Appendix I.

# 6 RELATED WORK

## 6.1 INSTRUCTIONS IN RETRIEVAL

The use of instructions is a relatively new development for IR models, as dense retriever training generally focuses on learning similarity functions similar to phrase-level matching (Craswell et al., 2020; Izacard et al., 2021; Wang et al., 2022a). Some of the earliest work on the topic is TART (Asai et al., 2022) and Instructor (Su et al., 2022) which used simple task prefixes during training. More recently, E5-Mistral (Wang et al., 2023), GritLM (Muennighoff et al., 2024), and NV-Retriever (de Souza P. Moreira et al., 2024) scaled up the dataset and model size. These newer models typically re-use the same instruction set proposed by the E5-Mistral model.[16] Our work differs from this by applying and evaluating adaptability *per-query* rather than using a dataset wide prefix.

Our robustness evaluation also goes beyond prior works: while, e.g., Su et al. (2022) tests instruction phrasing by changing one word, we consider a broader range of length/style modifications.

Several benchmark efforts focus on explicitly testing the instruction following ability of retrievers: FollowIR (Weller et al., 2024) and InstructIR (Oh et al., 2024). Both found that existing bi-encoder retrieval models fail to use instructions as an LM would. Our work presents the first bi-encoder that achieves significantly above-random performance on these benchmarks.

## 6.2 PROMPTING LMS

It is now the defacto-standard for LMs to take and reason over input instructions given via *prompting*. This was discovered and popularized by models such as InstructGPT (Ouyang et al., 2022), FLAN (Wei et al., 2022a), and T0 (Sanh et al., 2022). Importantly, these works found that diversity of training data was crucial to generalization. Instructions are also often included in LM's training data to encourage this behavior, both in pre-training data (Soldaini et al., 2024; Computer, 2023) and followed by stages of post-training (including fine-tuning/RL Groeneveld et al. (2024)).

Although IR models often use LMs as their base architecture before IR training, little work has explored using standard LM capabilities like promptability in IR. The closest works include GritLM (Muennighoff et al., 2024) who attempted to do in-context learning (ICL) with their trained retriever but found worse results than zero-shot and potentially the recent BGE-ICL,[17] although as of late September 2024, there is no associated paper describing their procedure or training details.

## 6.3 SYNTHETIC INSTRUCTION GENERATION

Generating synthetic data for training is a popular technique given the strong performance of LMs. This happens in both NLP (Ben Allal et al., 2024; Adler et al., 2024) as well as in IR, for generating queries or documents for training (Bonifacio et al., 2022; Dai et al., 2022; Jeronymo et al., 2023).

We build upon another line of work that uses LMs to create instructions that can be used for retrieval training (Wang et al., 2022b; Li et al., 2023a; Chung et al., 2024), however, we use this approach to generate instruction-negative passages instead of standard passages.

# 7 CONCLUSION

We presented the first zero-shot *promptable* retriever, Promptriever, trained from a new instruction-based retrieval dataset based on MS MARCO. Experiments show that Promptriever not only performs well on the standard retrieval task, but also follows instructions more effectively than prior work, adapting its notion of relevance per-query. Overall, this shows that techniques discovered in the LM community, such as prompting, can be extended to dense retrievers as well. We hope this will inspire joint research between the two communities and further enable retrievers that can adapt on the fly.

---

[16]You can find a list of all the "instructions" at this url.
[17]https://huggingface.co/BAAI/bge-en-icl

## 8 LIMITATIONS

Although Promptriever introduces per-instance prompting to retrieval models, there are many aspects of prompting with LMs that we did not explore in this work. For example, future work could look at in-context learning: can retrieval models be prompted with a few examples explicitly versus just imperative requests? We leave this to future work to explore and hope to see a wide variety of prompting techniques from the LM community applied to retrieval models.

We also note that, similar to language models, it is often unclear why some IR prompts perform better than others. Language models have become more robust to different prompts over time, and we hope that future work will continue to improve this ability for retrieval models.

Finally, as with any LM-generated data, it is possible that there are errors, pernicious social biases, and/or incorrect pieces of information in the generated passages and instructions. Although we applied some (probabilistic) correctness filters and conducted quantitative and qualitative explorations, it is still possible that unintended characteristics slipped through. While experiments demonstrate that training on our corpus improves performance, further audits of our corpus (and retrieval training sets more broadly) would be appropriate and useful.

## 9 ACKNOWLEDGMENTS

We thank the machine learning team at Samaya AI for the helpful discussions and feedback. OW is supported by a NSF GRFP fellowship.

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

## A    RETRIEVAL PROMPTS

In Table 10 we show the retrieval prompts and their scores per dataset in Table 11.

## B    CROSS-ENCODER VS BI-ENCODER EFFICIENCY

Cross-encoders compute full attention over combinations of queries and documents at inference time and are O(Q*D) where Q is the number of queries and D is the number of documents.

On the other hand, dense retrievers (a.k.a. bi-encoders) like Promptriever compute attention over queries and documents individually, and estimate relevance using a cheap dot product, yielding O(Q + D + dot-product). The dot product is negligible, and is often implemented with a vector database in practice.

Thus, we can see that cross-encoders are significantly more expensive to use, which is why most retrieval systems use a first stage pass with a bi-encoder and then follow that up with a cross-encoder.

## C    STATISTICAL SIMILARITY TESTS ON BEIR

Due to the complexity of the figure, we include statistical tests here. We compare two groups: Promptriever (no prompt) vs RepLLaMA (no prompt) and Promptriever (prompt) vs Promptriever (no promt).

(1) Promptriever vs RepLLaMA (no prompts): Promptriever is significantly better for Arguana, HotpotQA, Quora, SciDocs, while RepLLaMA is significantly better at NQ and Climate-Fever ($p<0.05$ for all), and they are statistically similar for the others.

(2) Promptriever with prompts vs without them: prompts are significantly better for Arguana, HotpotQA, Quora, SciDocs, and Climate-Fever ($p<0.05$) , while all others are statistically similar.

## D    INSTRUCTION DEFINITION

Most queries in retrieval are short and often ambiguous. In practice the crucial difference between a query and an instruction is that an instruction provides more specific input. This could include specifics for what defines relevance, specifics that define not-relevance, or any other background / additional information. Instructions are also typically longer than queries, although it is possible to have a short instruction or a long query. As both provide user input for what they are searching for, the exact boundary can be hard to define exactly. However, for current evaluation datasets, it fairly easy to disambiguate: queries from NQ and MSMARCO are standard and short, while those FollowIR and InstructIR provide many of the features defined earlier and are significantly longer.

## E    HYPERPARAMETER DETAILS

We use the following hyperparameters as given by the authors of the RepLLaMA on their Github page using Tevatron (Gao et al., 2022). This is using `meta-llama/Llama-2-7b-hf` with lora r 32, lora modules q_proj, k_proj, v_proj, o_proj, down_proj, up_proj, gate_proj, enabled bfloat16, using eos pooling, using normalization, a temperature of 0.01, learning rate of 1e-4, one epoch, passage length 256, 100 warm up steps, a train group size of 16, and an effective batch size of 128 (4 GPUs, 8 per device with a 4 accumulation steps). We differ from the original paper by using

query max length 304 to account for the longer instructions (previously set to 32 in RepLLaMA). Training takes approximately 2 days on 8x40GB A100s for the ablation runs and 4 days for the full run. Inference takes up to four hours per dataset on the 8x cluster using 512 length parameters for query and passages.

## F    GENERATION PROMPTS

We include the prompts used to generate the data. For the system prompts, we used Prompt 6, for the instruction generation we used Prompt 4, and for the instruction negatives we used Prompt 5.

## G    BERT-SIZE MODEL BACKBONES

We note that we also tried training BERT-sized backbones with the Promptriever data. However, we found significantly reduced performance (e.g. ~30 MRR on MS MARCO dev). We tried training with the exact same technique (LoRA, EOS token pooling) as well as more standard methods for BERT (CLS token, full fine-tuning) and had similar results. It is possible that these smaller models are not able to understand instructions as effectively, a finding which is mirrored in the LM community.

## H    MORE QUALITATIVE EXAMPLES

### H.1    FollowIR EXAMPLE

**Query:** Identify documents that discuss the reasons for automobile recalls.

**Instruction:** A relevant document will specify major or minor reasons for automobile recalls by car manufacturers. Documents that discuss truck recalls are not relevant.

RETRIEVED PASSAGES COMPARING PROMPTRIEVER AND REPLLAMA

**Promptriever:    Rank 1** (relevance=True):

YOUR WHEELS: DEFECT INVESTIGATIONS AT RECORD HIGH IN '89

February 1, 1990, Thursday, Home Edition The auto safety tally for 1989 has come in, and the news isn't particularly good for consumers. Last year, 6.7 million vehicles, including cars made by virtually every major producer, were recalled by manufacturers because of safety defects that needed correcting. The recalls included 1.7 million General Motors cars with sticking throttles, 1.4 million Ford Escorts with defective seats, 500,000 Chryslers with rusty front-suspension parts, and 408,000 Toyota Camrys with problem seat belts. The National Highway Traffic Safety Administration boasted that it opened 213 defect investigations during 1989, a record in the 20-year history of the agency. Of the 6.7 million recalls, the agency claims that it influenced 77%. The statistics beg the question: Are cars safer because more investigations are being run, or do cars have more defects than ever? The numbers, critics say, show the public is at risk...

**Rank 2** (relevance=False):

JAPAN WARNING ITS CAR MAKERS AFTER RASH OF DEFECTS, RE-CALLS

December 5, 1989, Tuesday, P.M. Final Transport Ministry officials will ask Japanese car makers to pay more attention to design and quality control because of defects found in new and remodeled cars recently, a ministry official said today. The officials will issue the request at a regularly scheduled meeting with the heads of the Japan Automobile Manufacturers Assn. on Thursday. Since April 1, 11 Japanese models and 14 foreign models have been recalled because of defects. This compares with 15 Japanese models and 24 foreign models in the year to March 31, 1989 ...

**RepLLaMA:** **Rank 1** (relevance=False):

> full-sized GM pickup trucks and cab-chassis equipped with fuel tanks mounted outboard of the frame rails contain a defect related to motor vehicle safety. At the same time, he released a comprehensive Engineering Analysis (EA) Report that explains in detail the basis for the decision. A summary of that Report is set out in the Federal Register notice announcing that decision. 59 Fed. Reg. 54025 (October 27, 1994). The EA Report makes two fundamental findings. First, on the basis of statistical analysis, laboratory crash tests, and a review of available information concerning specific real-world crashes, the EA Report demonstrates that the GM pickups at issue exhibit an increased risk of fire in side-impact crashes compared to similar vehicles (specifically, to contemporary Ford and Dodge full-sized pickups). Second, primarily on the . . .

**Rank 2** (relevance=False):

> or type of crash? If so, how relevant? 4. Are the extent of a manufacturer's awareness of a potential or ongoing safety risk, and the extent of a manufacturer's efforts to avoid that risk, relevant to the issue of whether an unreasonable risk exists? Is a manufacturer's failure to implement measures to mitigate or eliminate an increased safety risk relevant to that issue? 5. What weight should be given to the foregoing three factors, and any other relevant factors, in deciding whether a vehicle contains a defect related to motor vehicle safety? Other Information Sought 1. Additional information concerning post-crash fires in real-world crashes involving the GM vehicles covered by this investigation and comparable vehicles. 2. Additional information regarding GM's awareness of an increased risk of post-crash fires due to the placement of the fuel tanks outside of the frame rails of these trucks, both at the time of their design and afterward. . . .

**Comment:** You can see that RepLLaMA incorrectly focuses on trucks, despite the instruction to avoid them. Whereas Promptriever correctly focused on sedan recalls.

## H.2 INSTRUCTIR EXAMPLE

**Query:** what is the temperature in flagstaff az?

**Instruction:** As a remote worker considering relocating to Flagstaff, AZ, I'm trying to compare the temperature there with other cities on my list. My search would focus on a comprehensive weather tool that can show me a comparison of temperatures along with cost-of-living indices.

RETRIEVED PASSAGES COMPARING PROMPTRIEVER AND REPLLAMA

**Promptriever:** **Rank 1** (relevance=True):

> Comparison Tool for Climate and Living Expenses in Various Cities - February Insights Located at a higher elevation, this city experiences cooler temperatures in the winter months. The daytime highs around this time of year average at 7°C (45°F), while the nocturnal lows often plunge to an average of -10°C (14°F). This intuitive tool not only displays these temperature statistics for February but also allows you to juxtapose this data with that of other potential cities on your radar. Furthermore, for a comprehensive overview, this comparison can be tailored to include living expenses, presenting you with a dual-axis chart that informs both your budgeting and climate preferences.

**Rank 2** (relevance=False):

> Real estate stakeholders now have access to a comprehensive climate profile of Flagstaff, which includes an array of meticulously crafted weather charts. Each month is represented, showcasing temperature trends with peaks in July averaging a comfortable 78.2°F, while the winters introduce a cooler demeanor with December's average resting at 29.6°F. Precipitation peaks are highlighted, with October

Table 6: Word-level dataset statistics from augmenting MS MARCO train with instructions. The dataset was generated from a cross-product of length format and features with each subset having approximately 122k instances (and the total dataset ~490k). The mean numbers are rounded to the nearest digit. We see that Llama 3 70B generally followed the length description.

| | Category | Min | Mean | Max |
|---|---|---|---|---|
| Style Feature | None | 16 | 101 | 369 |
| | Negation | 18 | 98 | 363 |
| | Background | 19 | 108 | 340 |
| | Persona | 26 | 106 | 374 |
| Length Format | Short | 16 | 40 | 84 |
| | Medium | 43 | 84 | 154 |
| | Long | 43 | 107 | 185 |
| | Very Long | 96 | 181 | 374 |
| | All | 16 | 103 | 374 |

> notable for its rainfall, averaging approximately 1 inch. This climatological snapshot, accumulated from several years' data, caters to those placing importance on historical weather patterns. Additionally, curated for the environmentally conscious buyer, the health of Flagstaff's atmosphere is quantified, boasting lower air pollutant levels compared to state and national benchmarks. This information has been visualized in an easily digestible format, ready for integration in listings to guide clientele through the perks of Flagstaff's seasonal rhythms

**RepLLaMA:** Note that the relevant doc is not in the top 5.

**Rank 1** (relevance=False):

> Real estate stakeholders now have access to a comprehensive climate profile of Flagstaff ... (same as above)

**Rank 2** (relevance=False):

> During the past five years, Flagstaff, Arizona, has experienced varied climatic conditions pivotal for any adventurer planning an extended stay. In July, the warmest month, average highs have consistently hovered around 78 degrees, based on an in-depth analysis of the half-decade's climatic patterns. Conversely, the winter month of December has presented average lows approximating 30 degrees, a crucial detail for the correct selection of thermal gear. Historical monthly rainfall peaked in October, averaging 1 inch - a statistic that could influence the timing of hikes and outdoor activities. This data, sourced from a comprehensive review of meteorological records, ensures both the precision and reliability required for a meticulous travel guide. The air quality index and pollution levels, although not the primary concern for temperature planning, remain significantly better than statewide and national averages, thus assuring a breathable, pristine environment conducive to exploration.

**Comment:** RepLLaMA was not able to identify the most important piece of the instruction, the comprehensive weather tool. Promptriever identified it and placed it as rank 1 whereas it did not make the top 5 for RepLLaMA.

## I  GENERIC INSTRUCTIONS

We show the generic instructions given to the models in Table 7. These were generated by prompting GPT-4o and Claude-3.5-Sonnet for generic retrieval descriptions.

---

Prompt for Instruction Generation

## Input Data
I have the following query and REL_DOCS_NUM_FILL_ME documents which have been marked as relevant and NON_REL_DOCS_NUM_FILL_ME which are non-relevant.

Query: QUERY_FILL_ME

POS_DOC_FILL_ME

NEG_DOC_FILL_ME

## Your task
I need you to come up with an instruction that can be appended onto the end of this query that will **make only one relevant document** and **make all other documents (including previously relevant docs) non-relevant**. You can choose which document will stay relevant to the new instruction, by writing an instruction that applies to only one of the relevant documents (you choose). This additional instruction should provide a test for strong frontier language models to determine if they can follow instructions. Triple check your work to be certain that the chosen document is still relevant and that the others are non-relevant – if you mess up you will be fired. Do not give away the answer in the instruction!

For this example, please generate the instruction to be LENGTH_FORMAT_FILL_ME. **In the instructions, provide detailed specifics for what makes a document relevant.** Remember that this criteria should make the one document relevant and all others irrelevant. Also be sure that the **instruction is generic and does not contain the answer to the query**.

Output the response in JSON form only with no other text, with the keys, "instruction" (str), "relevant_docs" (one document id that is the first doc, e.g. "[2]") and "non-relevant_docs" (all other document ids, e.g. "[1,3,...]").

## **Your output (JSON only):**

---

Figure 4: Prompt for Instruction Generation

---

**Prompt for instruction negatives**

Generate three 100 word passages (and explanations) that directly answer the query but do not provide a valid document according to the specific query. Then generate one passage that matches both. Make it obvious to a reader which ones are which.

Query: QUERY_FILL_ME

Specific Query: INSTRUCTION_FILL_ME

Remember your goal is to **generate a relevant document (MS MARCO style, with passage and title) for the query** but a **non-relevant document for the specific query**. You should generate only factual information.

To be crystal clear, your generated documents should have related information about "QUERY_FILL_ME". However, these generated documents should not be relevant to the specific query. As examples, they may omit crucial information that is needed for the specific query, if the query is ambiguous it may use an alternative meaning, or it may specifically mention elements that are said to be non-relevant.

You should also generate an explanation with the category it is and a succinct reason. The tags are "different interpretation", "omission", "mention non-relevant flag" or "none" for the relevant to both. E.g. "omission - it does not mention [reason]". **Be sure the documents marked as non-relevant to both are actually not relevant to the specific query!!**

Remember! **It should be trivially obvious to a reader why they are non-relevant!**

Diverse Generated Documents in **JSON output** with "matches_both", "explanation", "title", and "passage" keys. Reply with only valid JSON, no other text:

---

Figure 5: Prompt for Instruction Negatives

---

**System Prompt for All Prompts**

You are an expert at writing precise detailed instructions for language models and are paid millions of dollars to be a data engineer for OpenAI. Your sole duty is to write instructions that can be used for training data for the next superpowerful model, GPT-6. Answer succinctly and carefully follow all instructions given so that you can earn your large bonus and not be fired.

---

Figure 6: System Prompt for All Prompts

## J  ERROR ANALYSIS ATTEMPTS

In order to further understand why Promptriever was more effective than RepLLaMA we conducted the following error analyses. For the datasets with the largest differences between the two (and for the difference between Promptriever with prompt and without prompt, such as Climate-FEVER, SciFact, and Arguana) we calculated the per-query nDCG@10 scores. We then binned the queries into those that saw improved performance vs those that did not. Finally, we fine-tuned a BERT-base model and a bag-of-words model on 80% of those examples, leaving 20% for a hold out test set. However, in every case we found that the accuracy was below the majority baseline (typically around 66%). The best AUC score we found for any dataset was 54%, further indicating that there was not much signal in the data. We attempted several other combinations (including adding queries with tied scores in the negative bin, adding the prompt text to the queries) none of which changed these results. We hypothesize that the documents must be a critical component to understanding why Promptriever works better or why the prompts are helpful (or alternatively, the query-document connection).

We further tried simple statistics of the binned queries but found the two groups were indentical w.r.t. length, idf, and other basic text statistics.

## K  COMPARISON TO STATE-OF-THE-ART MODELS ON BEIR

Table 12 compares Promptriever to some of the best models[18] (Muennighoff et al., 2024; Wang et al., 2023; Lee et al., 2024; BehnamGhader et al., 2024; Merrick et al., 2024; Li et al., 2023b) on the BEIR leaderboard (Muennighoff et al., 2022) (from MTEB). We note that our model does not train on any of the BEIR datasets except for MS MARCO, which puts it at a disadvantage (as most SOTA models use all training/dev sets). Despite this, we see solid performance, including middle-of-the-pack performance when compared on only datasets without train/dev sets (truly OOD performance) beating GritLM, LLM2Vec, and Google Gecko.

---

[18]Details for Voyage's model is here and TDTE here.

Table 7: Generated Generic Instructions for IR, as generated by GPT-4o and Claude-3.5-Sonnet in July 2024. Prompts asked the models to generate them of varying length.

| Generic Instructions |
| --- |
| • Retrieve relevant passages. |
| • Find answer-containing text. |
| • Rank based on relevance. |
| • Identify key information. |
| • Extract pertinent information. |
| • Rank documents based on query relevance. |
| • Retrieve passages that answer the user's question. |
| • Find relevant passages. |
| • Rank matching documents. |
| • Retrieve answer-containing text. |
| • Identify key information sources. |
| • Find and rank passages that best address the user's query. |
| • Locate text segments that are relevant to the query and rank them. |
| • Identify and retrieve passages that answer the user's question. |
| • Extract passages from the corpus that are most relevant to the given query. |
| • Rank passages based on their ability to address the core aspects of the query. |
| • Given a web search query, retrieve relevant passages that answer the query. |
| • Find relevant passages for the given query. |
| • Select the most relevant passages that directly answer this query. |
| • Rank documents based on their relevance and informativeness to the given question. |
| • Retrieve passages containing factual information that addresses this specific inquiry. |
| • Identify and rank sources that provide comprehensive answers to the posed question. |
| • Analyze the query to identify the key information needs. Retrieve and rank passages that provide comprehensive answers to those needs. |
| • Locate relevant passages that directly respond to the user's question. Ensure the passages are ranked based on their relevance and accuracy. |
| • Search for text that addresses the user's query. Rank the passages based on how well they meet the information needs and provide clear answers. |
| • Examine the query for specific details and retrieve passages that address those details. Rank the results by their relevance and comprehensiveness. |
| • Extract pertinent information from the corpus to address the given query. |
| • Locate and prioritize text segments that provide accurate answers to the user's question. |
| • Evaluate document relevance based on query similarity and information content. |
| • Identify passages containing key facts related to the input query. |
| • Parse the query, then retrieve and rank relevant textual information. |

Table 8: Examples of instruction features in our new dataset, including negation, a POV, and background information.

| Type | Example |
|------|---------|
| Negation | A relevant document is one that provides information about a specific city or town, including its location, population, and history. It should not be about a trail, a business, or a resort. The document should also contain specific details about the city or town, such as its county or state. **Documents that only mention the name of the city or town in passing are not relevant.** |
| Persona | **I'm a history teacher preparing a lesson on the origins of the Pledge of Allegiance** and I need documents that provide a clear and specific answer to when it was written, including the name of the author and their occupation. A relevant document should provide a direct quote or explicit statement about the creation of the Pledge. |
| Background | **In the field of chemistry, substances can be classified into different categories based on their composition and properties. A thorough understanding of these categories is essential to accurately identify and describe various substances.** When evaluating a document's relevance to the question of whether gasoline is a substance or mixture, consider the following criteria: a relevant document must explicitly address the composition of gasoline, discussing its homogeneity or heterogeneity, and provide specific details about its properties or behavior under different conditions. The document should also demonstrate a clear understanding of the distinction between substances and mixtures, and apply this understanding to the case of gasoline. Furthermore, a relevant document should not simply provide a general definition of a substance or mixture, but rather provide specific information about gasoline that helps to answer the question |

Table 9: Examples of instructions by length format.

| Type | Example |
|---|---|
| Short (1-2 sentences) | Documents that describe the authorship of a specific hymn or song, mentioning the writer's name and the song's title, are relevant. Documents that discuss general information about bands, poems, or biblical events are not relevant. |
| Medium (3-6 sentences) | Proton pump inhibitors are a class of medications that have been widely used for several decades. They are available both over-the-counter and by prescription. A relevant document should provide a clear explanation of how proton pump inhibitors work to reduce stomach acid, and specifically mention their effect on the body's production of stomach acid. The document should also discuss the medical conditions that proton pump inhibitors are used to treat. A relevant document should not simply list the names of proton pump inhibitors or their uses without providing a detailed explanation of their mechanism of action. |
| Long (one paragraph) | Nicotine is a highly addictive substance found in tobacco products, and its detection in the body is a crucial aspect of medical testing. The human body has various ways of eliminating nicotine, including through urine, blood, and hair follicles. When evaluating documents related to nicotine detection, it is essential to consider the specific context and criteria for relevance. A relevant document should provide a clear and concise answer to the question, specifying the duration of nicotine presence in the body, particularly in urine tests. The document should also discuss the factors that influence nicotine detection, such as the frequency and amount of smoking, as well as the role of passive smoking. Furthermore, a relevant document should provide a comprehensive overview of nicotine's effects on the body and its elimination process. Documents that merely list detection periods without providing a detailed explanation of the underlying factors or fail to address the specific context of urine tests should be considered non-relevant. |
| Very Long (two paragraphs) | I'm planning a road trip from Red Lodge to Cooke City, Montana, and I'm looking for information on the route that will take me through the most scenic and thrilling parts of the Montana-Wyoming border. I've heard that there's a particular highway that's known for its steep switchbacks and breathtaking views, and I want to know more about it. A relevant document would need to provide specific details about the highway, such as its name, elevation gain, and any notable features or landmarks along the way. It's crucial that the document focuses on the highway itself, rather than general information about road trips or travel in Montana and Wyoming. 

 I'm not interested in documents that talk about motorcycle helmets, natural arches, or teething symptoms - those are completely unrelated to my road trip plans. A relevant document should make me feel like I'm getting a firsthand account of the highway and its attractions.I've tried searching online, but I keep getting results that are either too vague or too focused on other aspects of travel. That's why I need a document that can provide me with the specific information I'm looking for. If a document can give me a clear sense of what to expect on this highway, including its length, elevation, and any notable features, then I'll know it's the right one. Anything less, and I'll have to keep searching. |

Table 10: Prompts used for BEIR experiments. Results for each dataset is shown in Table 11.

| Prompts |
| --- |
| • Be careful when assigning relevance as your job is on the line and I will give you a 1000 dollar tip. |
| • Think carefully about these conditions when determining relevance. |
| • A relevant document should also provide a clear and concise explanation, avoiding unnecessary complexity or ambiguity. When in doubt, prioritize documents that provide a clear, direct, and specific answer to the query. |
| • A document that meets these criteria is considered relevant, while a document that does not meet these criteria is considered non-relevant. |
| • A relevant document should focus solely on providing a clear and accurate answer to the query, without distracting or unnecessary information |
| • A document is relevant if it helps to answer the query. Surface relevant documents only. |
| • Relevant documents are those that are topically related, answer the given question, or otherwise provide insight on the input. Think step by step about whether a document is relevant for this question. |
| • Find relevant documents to the query. Use strict critera when evaluating relevance: a relevant document here should provide direct information to either fully answer the query, or provide useful information towards answering it. Avoid only topically relevant documents. |
| • When judging the relevance of a document, focus on the pragmatics of the query and consider irrelevant any documents for which the user would have used a different query. |
| • Think carefully about relevance |

Table 11: BEIR dataset scores for different prompts (shown larger in Table 10) for Promptriever

| Prompt | ARG | CFV | DBP | FEV | FQA | HQA | NFC | NQ | QUO | SCD | SCF | COV | TOU |
| --- | --- | --- | --- | --- | --- | --- | --- | --- | --- | --- | --- | --- | --- |
| A relevant document should focus solely on providing a clear and accurate answer to the query, without distracting or unnecessary information | 55.9 | 29.4 | 43.7 | 78.1 | 42.2 | 68.0 | 35.9 | 58.1 | **88.0** | 19.3 | 75.9 | 73.3 | 19.2 |
| A relevant document should also provide a clear and concise explanation, avoiding unnecessary complexity or ambiguity. When in doubt, prioritize documents that provide a clear, direct, and specific answer to the query. | **56.7** | **32.1** | 43.1 | 77.3 | 38.7 | 67.5 | 35.0 | 56.1 | 87.2 | **19.7** | 75.0 | 64.7 | 18.3 |
| Think carefully about relevance | 52.7 | 27.5 | 44.8 | **82.8** | 43.7 | **69.5** | 36.5 | 61.5 | 85.9 | 17.8 | 76.2 | 82.5 | **32.0** |
| A document that meets these criteria is considered relevant, while a document that does not meet these criteria is considered non-relevant. | 51.4 | 26.4 | **45.2** | 80.1 | **46.6** | 69.0 | **36.9** | **62.2** | 86.8 | 18.3 | 74.9 | **84.6** | 30.4 |
| Think carefully about these conditions when determining relevance. | 53.2 | 26.7 | 44.9 | 81.9 | 43.1 | 69.3 | 35.9 | 61.1 | 84.9 | 18.0 | **76.3** | 81.5 | 30.2 |
| When judging the relevance of a document, focus on the pragmatics of the query and consider irrelevant any documents for which the user would have used a different query. | 53.3 | 24.0 | 43.1 | 78.5 | 43.4 | 68.3 | 34.3 | 59.4 | 86.6 | 18.0 | 75.1 | 79.9 | 27.8 |
| Find relevant documents to the query. Use strict critera when evaluating relevance: a relevant document here should provide direct information to either fully answer the query, or provide useful information towards answering it. Avoid only topically relevant documents. | 51.5 | 26.2 | 44.4 | 79.3 | 45.3 | 67.3 | 36.6 | 60.0 | 87.2 | 18.5 | 75.2 | 82.4 | 30.4 |
| Relevant documents are those that are topically related, answer the given question, or otherwise provide insight on the input. Think step by step about whether a document is relevant for this question. | 54.6 | 27.0 | 43.8 | 80.9 | 41.2 | 68.9 | 35.4 | 57.2 | 86.6 | 17.6 | 76.1 | 77.9 | 24.1 |
| A document is relevant if it helps to answer the query. Surface relevant documents only. | 52.9 | 27.3 | 43.4 | 79.4 | 45.0 | 67.3 | 35.7 | 58.6 | 86.8 | 17.9 | 75.5 | 79.8 | 28.5 |
| Be careful when assigning relevance as your job is on the line and I will give you a 1000 dollar tip. | 52.4 | 24.3 | 43.2 | 81.0 | 41.0 | 68.6 | 35.5 | 60.5 | 84.4 | 18.3 | 75.4 | 83.6 | 25.4 |

Table 12: BEIR comparison for models in the MTEB leaderboard. Promptriever, unlike most others, has not been trained on the training/dev sets of the BEIR datasets (other than MS MARCO). Despite that, it performs comparably to many models on the true out-of-distribution (OOD) datasets that don't have train/dev sets.

| Dataset | GritLM-7B | e5-mistral-7b-instruct | voyage-lite-02-instruct | gte-Qwen1.5-7B-instruct | google-gecko | LLM2Vec-Mistral-supervised | snowflake-arctic-embed-l | Promptriever-llama2-7b | TDTE |
|---|---|---|---|---|---|---|---|---|---|
| **Has train/dev set** | | | | | | | | | |
| DBPedia | 46.6 | 48.9 | 39.8 | 48.0 | 47.1 | 49.6 | 46.0 | 45.2 | 53.2 |
| FEVER | 82.7 | 87.8 | 91.4 | 93.4 | 87.0 | 89.4 | 88.2 | 82.8 | 77.7 |
| FiQA2018 | 60.0 | 56.6 | 52.5 | 55.3 | 59.2 | 53.1 | 44.7 | 46.6 | 40.7 |
| HotpotQA | 79.4 | 75.7 | 75.5 | 72.3 | 71.3 | 74.1 | 75.2 | 69.5 | 41.3 |
| NFCorpus | 40.9 | 38.6 | 43.7 | 38.3 | 40.3 | 39.3 | 37.7 | 36.9 | 88.9 |
| NQ | 70.3 | 63.5 | 64.3 | 61.8 | 61.3 | 61.7 | 63.1 | 62.6 | 23.0 |
| Quora Retrieval | 89.5 | 89.6 | 87.6 | 89.6 | 88.2 | 87.8 | 87.4 | 88.8 | 79.6 |
| SciFact | 79.2 | 76.4 | 79.9 | 75.3 | 75.4 | 78.9 | 73.8 | 76.3 | 80.8 |
| **Doesn't have train/dev set** | | | | | | | | | |
| ArguAna | 63.2 | 61.9 | 70.3 | 62.7 | 62.2 | 57.5 | 59.1 | 56.7 | 49.5 |
| Climate FEVER | 30.9 | 38.4 | 32.0 | 44.0 | 33.2 | 35.2 | 39.3 | 32.1 | 49.0 |
| SCIDOCS | 24.4 | 16.3 | 20.2 | 27.7 | 20.3 | 22.5 | 21.4 | 19.7 | 25.2 |
| Touche2020 | 27.9 | 26.4 | 26.8 | 20.3 | 25.9 | 22.2 | 34.5 | 32.0 | 22.0 |
| TRECCOVID | 74.8 | 87.3 | 81.0 | 72.7 | 82.6 | 77.7 | 80.7 | 84.6 | 58.8 |
| Average | 59.2 | 59.0 | 58.8 | 58.6 | 58.0 | 57.6 | 57.8 | 56.4 | 53.1 |
| Average OOD | 44.3 | 46.0 | 46.1 | 45.5 | 44.8 | 43.0 | 47.0 | 45.0 | 47.0 |

Table 13: BEIR results for all models.

| Dataset | BM25 | | RepLLaMA | | Llama 2 | | Llama 3.1 Instruct | | Llama 3.1 | | Mistral v0.1 | |
|---|---|---|---|---|---|---|---|---|---|---|---|---|
| | No Prompt | Prompted | No Prompt | Prompted | No Prompt | Prompted | No Prompt | Prompted | No Prompt | Prompted | No Prompt | Prompted |
| Arguana | 36.6 | 36.4 | 48.6 | 49.0 | 51.8 | 56.7 | 54.3 | 58.9 | 54.2 | 57.0 | 51.9 | 58.0 |
| Climate-FEVER | 13.6 | 13.9 | 29.3 | 30.8 | 27.6 | 32.1 | 27.2 | 29.8 | 26.0 | 28.8 | 26.1 | 28.1 |
| DBPedia | 29.9 | 23.3 | 44.8 | 43.5 | 45.0 | 45.2 | 45.1 | 46.0 | 45.2 | 45.6 | 43.7 | 44.7 |
| FEVER | 48.1 | 45.3 | 82.9 | 85.3 | 82.8 | 82.8 | 83.5 | 85.5 | 82.8 | 84.5 | 80.2 | 81.8 |
| FiQA | 25.1 | 21.9 | 45.0 | 42.7 | 45.9 | 46.6 | 45.8 | 47.2 | 47.1 | 47.8 | 45.3 | 45.7 |
| HotpotQA | 56.9 | 54.6 | 68.8 | 67.9 | 69.2 | 69.5 | 70.9 | 71.7 | 70.5 | 71.4 | 68.8 | 69.6 |
| NFCorpus | 32.1 | 23.5 | 36.0 | 35.0 | 36.5 | 36.9 | 37.7 | 38.5 | 37.6 | 37.6 | 36.5 | 37.0 |
| NQ | 28.5 | 25.4 | 63.0 | 62.2 | 61.9 | 62.6 | 62.3 | 63.8 | 62.7 | 63.6 | 62.1 | 62.6 |
| Quora | 80.4 | 75.3 | 86.0 | 85.5 | 86.5 | 88.0 | 86.2 | 87.3 | 83.6 | 85.4 | 84.4 | 85.1 |
| SCIDOCS | 15.8 | 14.9 | 16.1 | 16.7 | 17.3 | 19.7 | 18.4 | 20.8 | 18.1 | 20.6 | 17.6 | 19.8 |
| SciFact | 68.7 | 65.7 | 75.3 | 75.7 | 75.0 | 76.3 | 74.6 | 77.5 | 74.3 | 76.8 | 75.8 | 76.9 |
| TREC-COVID | 62.3 | 35.9 | 83.9 | 82.7 | 83.9 | 84.6 | 83.1 | 84.5 | 82.3 | 82.8 | 83.8 | 83.0 |
| Touche-2020 | 33.1 | 30.8 | 34.1 | 35.9 | 31.4 | 32.0 | 32.5 | 31.7 | 32.3 | 32.1 | 30.5 | 31.4 |
| Average | 40.9 | 35.9 | 54.9 | 54.8 | 55.0 | 56.4 | 55.5 | 57.2 | 55.1 | 56.5 | 54.4 | 55.7 |

