# OpenReview forum: "Promptriever: Instruction-Trained Retrievers Can Be Prompted Like Language Models"
_ICLR.cc/2025/Conference — ICLR 2025 Poster_

### Official Review · Reviewer_89Cp · 2024-10-26

**Soundness:** 3
**Presentation:** 4
**Contribution:** 3
**Rating:** 5
**Confidence:** 4

**Summary:**

The paper presents Promptriever, an innovative retrieval model that incorporates instruction-following capabilities akin to LLMs. By curating a new instruction training set from MS MARCO, the authors developed a bi-encoder retriever that adjusts its responses based on natural language instructions, demonstrating substantial improvements in retrieval tasks. Specifically, Promptriever achieves state-of-the-art results on instruction-following retrieval tasks, showing significant robustness to query phrasing and the ability to perform hyperparameter search via prompting. This model sets a new precedent for future work in aligning LLM prompting techniques with IR systems.

**Strengths:**

- The focus on enabling retrievers to understand human instructions aligns well with the current demand for more intuitive and adaptable search technologies, marking a significant advancement in retrieval systems.
- The methodological approach of not requiring human annotations is notable, offering a scalable and cost-effective solution for training retrieval systems.
- The clarity of the paper and its comprehensive details support replicability and transparency, which are crucial for advancing research in this area.

**Weaknesses:**

My main concern is about the **experiments**.
- The t-test is not conducted on the experimental results. In IR, it is very important to validate whether the improvement is significant.
- The use of varying amounts of training data for different models (Promptriever vs. RepLLaMA) raises concerns about the fairness and validity of the comparative performance analysis.
- The performance improvement on in-domain retrieval is very marginal. I guess the instruction-tuning process may affect the original retrieval performance.
- In Table 5, the selected baselines are not strong on BEIR benchmark. For example, E5-mistral has achieved 56.9 on BEIR. I think the authors should compare their methods with more advanced baselines.
- A minor suggestion: please use `` and ‘’ (two single quotes) to generate double quotes in LaTeX.

**Questions:**

Please consider to address my concerns in Weaknesses.

---

> ### Author Response · Authors · 2024-11-20
>
> We thank the reviewer for noticing that our work "mark[s] a significant advancement in retrieval systems" and offers a "scalable and cost-effective solution"!
>
> > The use of varying amounts of training data for different models (Promptriever vs. RepLLaMA) raises concerns about the fairness and validity of the comparative performance analysis.
>
> **We show ablations with the same training data in Table 4**, for our models except our final Joint version which combines all the instruction and standard data. We see that even with the same amount of training data (e.g. the “w/Instruction Negatives row”) there is a large gain in performance on instructions: +11.4 p-MRR, +3.2 nDCG@10 on FollowIR, +5.8 nDCG@10 and +13.1 Robustness@10 on InstructIR. As shown in our t-tests, we see these differences are statistically significant also.
>
> > The performance improvement on in-domain retrieval is very marginal. I guess the instruction-tuning process may affect the original retrieval performance.
>
> We agree that we do not show gains on in-domain data without instructions (as stated on lines 250-251).
>
> However, we show comparable performance with no instruction-based evals (e.g. it doesn’t hurt to add instructions) **while also showing significant improvements** on data that does have prompts/instructions (Tables 1 and 3).
>
> As you say, it is intuitive that adding instruction-based training data doesn’t help the model more than normal when it’s not given instructions/prompts at test time. However, as we show, adding instructions does enable new abilities when they are given.
>
> > In Table 5, the selected baselines are not strong on BEIR benchmark. For example, E5-mistral has achieved 56.9 on BEIR. I think the authors should compare their methods with more advanced baselines.
>
> **We do include E5-Mistral in Table 1** for instructions, as well as in Table 12 in the appendix on BEIR.
>
> As E5-Mistral was trained on significantly more data than just MS MARCO, including the training sets of BEIR, we cannot equally compare. BEIR was designed to be a zero-shot test set, and when models train on the training sets, we are unable to equally compare their performance as it is no longer zero-shot for them..
>
> However, we show in Table 12 that on the few datasets in BEIR that are truly out of domain (no train data) that we score similarly to E5-Mistral despite training with much less data. This showcases the usefulness of instruction-training.
>
> > A minor suggestion: please use `` and ‘’ (two single quotes) to generate double quotes in LaTeX.
>
> Thank you for the suggestion, we have fixed this.

---

> > ### Author Response · Authors · 2024-11-20
> > **Statistical Significant Tests**
> >
> > > The t-test is not conducted on the experimental results. In IR, it is very important to validate whether the improvement is significant.
> >
> > Thank you for the suggestion! We have performed the requested tests and will add them to the paper. After performing these tests, **our conclusions are the same as stated in the paper: significant improvements (p<0.05 for many cases) on data with prompts/instructions and similar performance on data without instructions/prompts.**
> >
> > Details are as follows: We use a two sided student t-test with p<0.05 significance for nDCG and MAP and use a Wilcoxon signed rank non-parametric test (due to a non-normal distribution) for p-MRR and Robustness@10.  As a brief summary:
> >
> > **Table 1:**
> >
> > Focusing on Bi-Encoders (since our contribution is Promptriever):
> >
> > FollowIR: in nDCG, Promptriever is significantly better (p<0.05) than all others in Robust04 and statistically similar to the best (Google Gecko) in Core17 and News21. For p-MRR, Promptriever is significantly better than all others in Robust04 and Core17 (p<0.05), and all others in News21 except similar to Google Gecko.
> >
> > InstructIR: Promptriever is significantly better (p<0.05) than all others in both metrics.
> >
> > **Table 2:**
> >
> > On MS MARCO Dev RepLLaMA is statistically better (p<0.05), while for DL19 and DL20 RepLLaMA and Promptriever are statistically similar. This similarity is expected and stated in the paper (Section 4.2).
> >
> > **Table 3**
> >
> > Promptriever vs RepLLaMA (no prompts): Promptriever is significantly better for Arguana, HotpotQA, Quora, SciDocs, while RepLLaMA is significantly better at NQ and Climate-Fever (p<0.05 for all), and they are statistically similar for the others.
> >
> > Promptriever with prompts vs without them: prompts are significantly better for Arguana, HotpotQA, Quora, SciDocs, and Climate-Fever (p<0.05) , while all others are statistically similar.
> >
> > **Table 4:**
> >
> > FollowIR: Promptriever (Joint) is better in p-MRR than all other models for all datasets (p<0.05), except similar to w/Hard Negatives for Robust04 and News21 and w/Instructions for News21. nDCG scores are statistically similar for all models on News21 and Core17, and for Robust04 w/Hard Negatives, w/Instructions, and Joint are statistically similar and better than all others (p<0.05).
> >
> > InstructIR: Joint, w/Instruction Negatives, and w/Instructions are statistically similar and better than all others in Robustness@10 and Joint and w/Instructions are statistically better than all others in nDCG (p<0.05 for all).

---

> > > ### Comment · Reviewer_89Cp · 2024-11-24
> > > **Reply**
> > >
> > > Thank you for your response!
> > >
> > > The t-test results you provided have indeed enhanced the reliability of the findings, addressing my concern 1 effectively.
> > >
> > > However, I maintain that the observed performance improvements are likely attributable to the training data being supplemented with instructions. This approach appears to yield a noticeable advantage over models that were not trained similarly. Therefore, the comparison is not very "fair".
> > >
> > > Additionally, I am interested in understanding the specific benefits of incorporating instructions directly into queries when compared to traditional techniques such as query rewriting or decomposition. Could you elaborate on how this method might offer superior results?

---

> > > > ### Author Response · Authors · 2024-11-24
> > > > **Query re-writing**
> > > >
> > > > > Additionally, I am interested in understanding the specific benefits of incorporating instructions directly into queries when compared to traditional techniques such as query rewriting or decomposition. Could you elaborate on how this method might offer superior results?
> > > >
> > > > Great question! Query rewriting/decomposition is a separate and complementary approach that could be used with or without instructions. For example, one could use decompositions for queries with instructions as they will natural decompose better (e.g. more complex information that will need to be decomposed).
> > > >
> > > > As it is complementary, we leave further exploration of query rewriting with instruction based retrievers to future work.  Our focus is solely on the benefits that instructions bring retrievers without additional outside help.

---

> ### Author Response · Authors · 2024-11-24
> **data comparison**
>
> Thank you for the response!
>
> > However, I maintain that the observed performance improvements are likely attributable to the training data being supplemented with instructions. This approach appears to yield a noticeable advantage over models that were not trained similarly. Therefore, the comparison is not very "fair".
>
> The main contribution of our work is indeed the careful, scaled data curation process, which (as you noted) provides a “noticeable advantage”. Our contribution is not a new neural architecture. The comparisons we made (including the data volume ablation --- which shows that training with our instruction corpus, and not just training with more data, indeed causes performance improvements) are intended to illustrate the advantages of our data curation method.
>
> We are the first to propose using query-specific instructions data for bi-encoders in IR and we do indeed show it provides advantages (but only if done carefully, as we describe and ablate). Many works curate new training sets as the main contribution, as we also do.
>
> If you still feel this is unfair, would you be willing to elaborate further?

---

> > ### Author Response · Authors · 2024-12-01
> >
> > As the discussion period ends soon, we wanted to ask if our response answered your concern?
> >
> > We have tried as much as possible to make a fair comparison by comparing to both SOTA models (Tables 1 and 12) while also showing ablations that keep the exact same training recipe, differing only on our novel instruction approach (Table 4).

---

### Official Review · Reviewer_M1QE · 2024-11-06

**Soundness:** 4
**Presentation:** 3
**Contribution:** 3
**Rating:** 8
**Confidence:** 4

**Summary:**

The paper introduces Promptriever, a bi-encoder model that enriches the per-instance query context within prompts. To train this model, per-query instructions and instruction negatives are synthetically generated. The authors evaluate Promptriever across various scenarios, including instruction-following datasets, in-domain retrieval, and out-of-domain retrieval. The model not only surpasses previous bi-encoders but also achieves performance comparable to state-of-the-art cross-encoder retrievers, showcasing its effectiveness in handling free-form language prompts.

**Strengths:**

- The paper introduces a novel idea by challenging the assumption that the same instruction can be applied uniformly across queries.
- The authors curate a training dataset that serves as a solid resource for providing detailed, per-instance instructions.
- I particularly liked the analysis in Section 5, which addressed several questions I had while reading the methodology and experimental results sections.

**Weaknesses:**

- The generated instructions are only used during model training. It would have been insightful to hold out a dev or test split to evaluate how the model performs on this dataset.
- For BeIR, the use of generic instructions feels too broad. Task-specific instructions, such as those in TART, might have been more effective and better aligned with the goals of the paper.

**Questions:**

- For out-of-domain retrieval, what is the rationale behind selecting the best score out of 10 prompts, rather than fixing a prompt or reporting an average score?
- Could the authors provide some qualitative examples illustrating how Promptriever performs across different categories in Table 6?
- In Table 11, simpler prompts like “Think carefully about relevance” seem to yield strong performance. Why might these simpler prompts outperform other more detailed ones that closely match the training data?

---

> ### Author Response · Authors · 2024-11-20
>
> We thank the reviewer for their time and for their feedback that our work provides a "novel idea" and a "solid resource"!
>
> > For out-of-domain retrieval, what is the rationale behind selecting the best score out of 10 prompts, rather than fixing a prompt or reporting an average score?
>
> In the first half of Table 4, we report results in a train/dev/test setting, selecting the prompt for each dataset that works best on the dev set (then reporting on the dev set). This standard ML setup simulates how a system might be used in practice.
>
> We agree that the options you mentioned would also be interesting+valid; however, they would underestimate the model. The reason we chose the train/dev/test method is: 1) to show that you *can* reliably improve retrieval performance, just with prompting (like with LLMs: practitioners generally optimize a prompt for each task/dataset [1] using a dev set); and 2) to illustrate that there is no “magic bullet” prompting retrieval models on all datasets/tasks (e.g., even “let’s think step by step” only works for math and logic problems for LLMs [2]). This thus aligns IR with ML/NLP and simulates the setting that would happen in practice if someone was trying to apply Promptriever to new out of domain data.
>
> References:
>
> [1] “DSPy: Compiling Declarative Language Model Calls into Self-Improving Pipelines” by Khattab et. al. 2024
>
> [2] “To CoT or not to CoT? Chain-of-thought helps mainly on math and symbolic reasoning” by Sprague et. al. 2024
>
> > In Table 11, simpler prompts like “Think carefully about relevance” seem to yield strong performance. Why might these simpler prompts outperform other more detailed ones that closely match the training data?
>
> This is a great question! We agree that we were also unable to find any correlation between score and prompt. **It seems, similar to LLMs, that prompting has various quirks that may be dependent on the model and data** (see Limitation, lines 475-478). Unfortunately, as this is still an issue with LMs, it seems like it will carry over into IR as well.
>
> > Could the authors provide some qualitative examples illustrating how Promptriever performs across different categories in Table 6?
>
> Unfortunately the examples in Table 6 are synthetic and are used for contrastive training. Thus, the only examples we could show are that the model contrastively prefers one over the other (there is no test set, just one positive and one negative document per instruction for training).
>
> We would be happy to include more qualitative examples from the test sets however. We show one example from FollowIR and one from InstructIR and show it in the general comment section.

---

> > ### Comment · Reviewer_M1QE · 2024-11-25
> > **Reply**
> >
> > Thank you for the reply.
> > It would have been great to have an explanation for why simpler prompts work better, but I understand this is not the scope of the current study.
> > I think my score is high enough, so I will maintain it as is.

---

### Official Review · Reviewer_CGMT · 2024-11-07

**Soundness:** 3
**Presentation:** 3
**Contribution:** 3
**Rating:** 6
**Confidence:** 4

**Summary:**

In this paper, the authors introduce an instruction-aware retriever named Promptriever, which can be prompted like an LM. They build a dataset for training such a retriever based on MS MARCO. They conduct experiments based on the dataset, and reveal several interesting conclusions.


Pros:
- The problem discussed in the paper, i.e., building instruction-following IR models, is interesting.
- The problem definition is clear. The proposed method is easy to follow.
- Experimental results verify the effectiveness of the proposed method.


Cons:
 - The dataset is generated solely based on MS MARCO. It would be great if more datasets could be considered, especially for the test. BEIR should also be considered for the zero-shot scenario.
- It would be great if the definition of instructions in IR could be clearly defined. For example, given the example "Which type of volcano eruption has not been seen?", the authors claim that the volcano types and formation can be added as additional instructions. I wonder why these are treated as additional instructions but not part of the original information need (i.e., the original queries).
- More details about the experimental setup should be provided. For example, are the baseline models trained using similar assessment data (with instructions in the queries)?
- Better baseline models should be added. For example, we can generate query rewrites via LLMs and then retrieve documents using the rewritten query (converting the NLP-like instruction into keyword-like queries).

**Strengths:**

Pros:
- The problem discussed in the paper, i.e., building instruction-following IR models, is interesting.
- The problem definition is clear. The proposed method is easy to follow.
- Experimental results verify the effectiveness of the proposed method.

**Weaknesses:**

Cons:
 - The dataset is generated solely based on MS MARCO. It would be great if more datasets could be considered, especially for the test. BEIR should also be considered for the zero-shot scenario.
- It would be great if the definition of instructions in IR could be clearly defined. For example, given the example "Which type of volcano eruption has not been seen?", the authors claim that the volcano types and formations can be added as additional instructions. I wonder why these are treated as additional instructions but not part of the original information need (i.e., the original queries).
- More details about the experimental setup should be provided. For example, are the baseline models trained using similar assessment data (with instructions in the queries)?
- Better baseline models should be added. For example, we can generate query rewrites via LLMs and then retrieve results using the rewritten query (converting the NLP-like instruction into keyword-like queries).

**Questions:**

- How are the baseline models trained? Are they using similar assessment data (with instructions in the queries)?

---

> ### Author Response · Authors · 2024-11-20
>
> We thank the reviewer for their time and for noticing that our work has a "clear" problem definition with an "easy to follow" and "effective" method!
>
> > The dataset is generated solely based on MS MARCO. It would be great if more datasets could be considered, especially for the test. BEIR should also be considered for the zero-shot scenario.
>
> **We do evaluate on BEIR in Table 3** and have a comparison to more models in Table 12 in the appendix.
>
> BEIR is designed to be a zero-shot test set, meaning that they do not want authors to train on the training sets, so we do not train on them. In Table 12 we compare to models which did train on the training sets, but show on datasets without training sets (e.g. truly OOD) that we approach SOTA.
>
> In fact, this showcases the strong performance of Promptriever’s training recipe, as we achieve comparable results while only focusing on MS MARCO for training – without training on the millions of other examples (and domains) used by other models.
>
> > It would be great if the definition of instructions in IR could be clearly defined. For example, given the example "Which type of volcano eruption has not been seen?", the authors claim that the volcano types and formations can be added as additional instructions. I wonder why these are treated as additional instructions but not part of the original information need (i.e., the original queries).
>
> Typically in IR a query is short string, to the point, and often ambiguous. We evaluate on many standard+popular text IR datasets like MSMarco, BEIR, TREC, etc that define queries this way.
>
> FollowIR and InstructIR (our other evaluation corpora) make queries more precise with instructions. An instruction could be: specifics for what defines relevance, specifics that define not-relevance, or any other background / additional information.
>
> We will add this definition and discussion in revision, thank you.
>
> > More details about the experimental setup should be provided. For example, are the baseline models trained using similar assessment data (with instructions in the queries)?
>
> The baselines in Table 4 do train with instructions and are the ablations for how we picked our recipe. All of the models except the final “Joint” model have the same data, as the Joint model fully combines the full instruction and non-instruction dataset, whereas the others split it (lines 355-357).
>
> For all the other results, the baselines are not trained with similar data. **This is the main contribution of Promptriever: we are the first to suggest using query-level instruction data in dense retriever training.** We show the difference these instructions make by directly using RepLLaMA’s recipe but adding instruction data.
>
> > Better baseline models should be added. For example, we can generate query rewrites via LLMs and then retrieve results using the rewritten query (converting the NLP-like instruction into keyword-like queries).
>
> We agree that one could have an LLM re-write a query and search with that (called query expansion). Query expansion is well-studied and has been shown to help retrieval models generally. As such, it is orthogonal to our work as these query expansions could be applied to instruction-based retrievers also.
>
> As our paper is focused on comparing the retrieval models themselves, we leave the analysis of the impact of query-expansions and their interactions with other LMs to future work.

---

> > ### Comment · Reviewer_CGMT · 2024-11-26
> >
> > Thank you for your reply, especially for the experiments with BEIR.

---

> > > ### Author Response · Authors · 2024-12-01
> > >
> > > > Thank you for your reply, especially for the experiments with BEIR.
> > >
> > > Thank you for the response! If our BEIR experiments addressed your concern, would you consider increasing your score?

---

### Official Review · Reviewer_UmzZ · 2024-11-08

**Soundness:** 3
**Presentation:** 3
**Contribution:** 3
**Rating:** 6
**Confidence:** 3

**Summary:**

Currently, retrievers are only able to retrieve texts similar to input queries, mostly with text similarity. In this paper, the authors present Promptriever, the first retrieval model able to be prompted with textual instructions.

For example, the users can pass in complex instructions to filter the passages to be relevant to a specific topic or exclude certain categories of passages.

To do so, the authors create and release a synthetic dataset of query-passage relevance pairs augmented with instructions. They use the MS MACRO dataset and generate instructions using Llama-3-70B, which includes diverse length formats and styles. Then, they use GPT-4o to generate the instruction negative passages. The Promptriever is then trained on the augmented data.

As a result, the Promptriever maintains strong retrieval scores in standard settings. Compared to the original RepLlaMA, it also follows instructions better. The authors also perform multiple ablations such as the instruction-negatives.

**Strengths:**

1. The motivation is strong and the story is convincing. The authors identify the retrievers' current lack of instruction-following abilities to motivate the method. They help to bridge the gap by introducing Promptriever.
2. The experiment performance is promising and shows improvements in various datasets.
3. The ablation studies are sufficient and necessitate the needs of the different training components.

**Weaknesses:**

See questions.

**Questions:**

1. Can you list out some use cases where the promptriever is used in scenarios such as RAG? How would it perform?
2. It would be nice to see some qualitative examples of Promptriever compared to traditional retrievers, besides the example in the intro.
3. You mentioned in L240 that cross-encoders perform best due to their significant compute advantage. Could you list out the compute resources required by different baselines and Promptriever?

---

> ### Author Response · Authors · 2024-11-20
>
> We thank the reviewer for their time and for noticing that our paper's "motivation is strong and the story is convincing"!
>
> > Can you list out some use cases where the promptriever is used in scenarios such as RAG? How would it perform?
>
> This is a great question - RAG crucially depends on the results gathered from the retrieval model. Promptriever, like any other IR model, could be used for RAG. The ability to prompt the retrieval model could add additional benefits to RAG, such as less reliance on keywords, ability to pass along “big picture” information, and/or additional instructions to enable RAG for very complex topics.
>
> **However, this paper is focused only on the task of retrieval itself.** As such, we leave other applications of promptable retrievers, such as RAG, to future work.
>
> > You mentioned in L240 that cross-encoders perform best due to their significant compute advantage. Could you list out the compute resources required by different baselines and Promptriever?
>
> Cross-encoders compute full attention over **combinations** of queries and documents at inference time and are O(Q*D) where Q is the number of queries and D is the number of documents. On the other hand, dense retrievers (a.k.a. bi-encoders) like Promptriever compute attention over queries and documents individually, and estimate relevance using a cheap dot product, yielding O(Q + D + dot-product) (the dot product is negligible, and is often implemented with a vector database in practice). We will add this discussion in revision.
>
> As for the comparison between other dense retrieval baselines, the difference will be the same as your normal intuitions: RepLLaMA and Promptriever use 7B models (e.g. LLaMA) while other models like TART use BERT. The cost to encode all documents depends on the base model (e.g. Llama vs BERT).
>
> > It would be nice to see some qualitative examples of Promptriever compared to traditional retrievers, besides the example in the intro.
>
> Thank you for the suggestion! We have added some qualitative examples of the various categories in the general section and will add them to the appendix. We show one from FollowIR and one from InstructIR.

---

> > ### Comment · Reviewer_UmzZ · 2024-11-25
> > **Reply**
> >
> > Thank you for the reply. I'd like to maintain my original score.

---

### Author Response · Authors · 2024-11-20
**Qualitative Example from FollowIR**

Several reviewers requested additional qualitative examples. We will include these in the appendix of our paper.

---------

## FollowIR Example

**Query:** Identify documents that discuss the reasons for automobile recalls.

**Instruction:** A relevant document will specify major or minor reasons for automobile recalls by car manufacturers. Documents that discuss truck recalls are not relevant.

**Retrieved Passages comparing Promptriever and RepLLaMA**

### Promptriever:

Rank 1 (relevance=True): YOUR WHEELS: DEFECT INVESTIGATIONS AT RECORD HIGH IN '89 \n\n February 1, 1990, Thursday, Home Edition The auto safety tally for 1989 has come in, and the news isn't particularly good for consumers. Last year, 6.7 million vehicles, including cars made by virtually every major producer, were recalled by manufacturers because of safety defects that needed correcting. The recalls included 1.7 million General Motors cars with sticking throttles, 1.4 million Ford Escorts with defective seats, 500,000 Chryslers with rusty front-suspension parts, and 408,000 Toyota Camrys with problem seat belts. The National Highway Traffic Safety Administration boasted that it opened 213 defect investigations during 1989, a record in the 20-year history of the agency. Of the 6.7 million recalls, the agency claims that it influenced 77%. The statistics beg the question: Are cars safer because more investigations are being run, or do cars have more defects than ever? The numbers, critics say, show the public is at risk….

Rank 2 (relevance=False): JAPAN WARNING ITS CAR MAKERS AFTER RASH OF DEFECTS, RECALLS \n\n December 5, 1989, Tuesday, P.M. Final Transport Ministry officials will ask Japanese car makers to pay more attention to design and quality control because of defects found in new and remodeled cars recently, a ministry official said today. The officials will issue the request at a regularly scheduled meeting with the heads of the Japan Automobile Manufacturers Assn. on Thursday. Since April 1, 11 Japanese models and 14 foreign models have been recalled because of defects. This compares with 15 Japanese models and 24 foreign models in the year to March 31, 1989 …

### RepLLaMA:

Rank 1 (relevance=False): full-sized GM pickup trucks and cab-chassis equipped with fuel tanks mounted outboard of the frame rails contain a defect related to motor vehicle safety. At the same time, he released a comprehensive Engineering Analysis (EA) Report that explains in detail the basis for the decision. A summary of that Report is set out in the Federal Register notice announcing that decision. 59 Fed. Reg. 54025 (October 27, 1994). The EA Report makes two fundamental findings. First, on the basis of statistical analysis, laboratory crash tests, and a review of available information concerning specific real- world crashes, the EA Report demonstrates that the GM pickups at issue exhibit an increased risk of fire in side-impact crashes compared to similar vehicles (specifically, to contemporary Ford and Dodge full-sized pickups). Second, primarily on the …

Rank 2 (relevance=False): or type of crash? If so, how relevant? 4. Are the extent of a manufacturer's awareness of a potential or ongoing safety risk, and the extent of a manufacturer's efforts to avoid that risk, relevant to the issue of whether an unreasonable risk exists? Is a manufacturer's failure to implement measures to mitigate or eliminate an increased safety risk relevant to that issue? 5. What weight should be given to the foregoing three factors, and any other relevant factors, in deciding whether a vehicle contains a defect related to motor vehicle safety? Other Information Sought 1. Additional information concerning post-crash fires in real-world crashes involving the GM vehicles covered by this investigation and comparable vehicles. 2. Additional information regarding GM's awareness of an increased risk of post-crash fires due to the placement of the fuel tanks outside of the frame rails of these trucks, both at the time of their design and afterward. ...

### **Our Comment**
You can see that RepLLaMA incorrectly focuses on trucks, despite the instruction to avoid them. Whereas Promptriever correctly focused on sedan recalls.

---

> ### Author Response · Authors · 2024-11-20
> **Qualitative Example from InstructIR**
>
> ## InstructIR Example
>
> **Query:** what is the temperature in flagstaff az?
>
> **Instruction:** As a remote worker considering relocating to Flagstaff, AZ, I'm trying to compare the temperature there with other cities on my list. My search would focus on a comprehensive weather tool that can show me a comparison of temperatures along with cost-of-living indices.
>
> **Retrieved Passages comparing Promptriever and RepLLaMA**
>
> ### Promptriever:
> Rank 1 (relevance=True): Comparison Tool for Climate and Living Expenses in Various Cities - February Insights Located at a higher elevation, this city experiences cooler temperatures in the winter months. The daytime highs around this time of year average at 7\u00b0C (45\u00b0F), while the nocturnal lows often plunge to an average of -10\u00b0C (14\u00b0F). This intuitive tool not only displays these temperature statistics for February but also allows you to juxtapose this data with that of other potential cities on your radar. Furthermore, for a comprehensive overview, this comparison can be tailored to include living expenses, presenting you with a dual-axis chart that informs both your budgeting and climate preferences.
>
> Rank 2 (relevance=False): Real estate stakeholders now have access to a comprehensive climate profile of Flagstaff, which includes an array of meticulously crafted weather charts. Each month is represented, showcasing temperature trends with peaks in July averaging a comfortable 78.2\u00b0F, while the winters introduce a cooler demeanor with December's average resting at 29.6\u00b0F. Precipitation peaks are highlighted, with October notable for its rainfall, averaging approximately 1 inch. This climatological snapshot, accumulated from several years' data, caters to those placing importance on historical weather patterns. Additionally, curated for the environmentally conscious buyer, the health of Flagstaff's atmosphere is quantified, boasting lower air pollutant levels compared to state and national benchmarks. This information has been visualized in an easily digestible format, ready for integration in listings to guide clientele through the perks of Flagstaff's seasonal rhythms
>
> ### RepLLaMA (note that the relevant doc is not in the top 5):
>
> Rank 1 (relevance=False): Real estate stakeholders now have access to a comprehensive climate profile of Flagstaff … (same as above)
>
> Rank 2 (relevance=False): During the past five years, Flagstaff, Arizona, has experienced varied climatic conditions pivotal for any adventurer planning an extended stay. In July, the warmest month, average highs have consistently hovered around 78 degrees, based on an in-depth analysis of the half-decade\u2019s climatic patterns. Conversely, the winter month of December has presented average lows approximating 30 degrees, a crucial detail for the correct selection of thermal gear. Historical monthly rainfall peaked in October, averaging 1 inch - a statistic that could influence the timing of hikes and outdoor activities. This data, sourced from a comprehensive review of meteorological records, ensures both the precision and reliability required for a meticulous travel guide. The air quality index and pollution levels, although not the primary concern for temperature planning, remain significantly better than statewide and national averages, thus assuring a breathable, pristine environment conducive to exploration.
>
> ### **Our Comment:**
>  RepLLaMA was not able to identify the most important piece of the instruction, the comprehensive weather tool. Promptriever identified it and placed it as rank 1 whereas it did not make the top 5 for RepLLaMA.

---

### Meta-Review · Area_Chair_PdCG · 2024-12-20

**Metareview:**

This paper introduces Promptriever, a retrieval model that can be prompted like a language model. The authors construct an instance-level instruction training set from MS MARCO. Experimental results show that Promptriever performs well on both standard retrieval tasks and instruction-following tasks. Extensive analyses are also provided.

Strengths:
1. The idea is well-motivated and interesting.
2. The authors construct a training dataset that can be a good resource for this problem.
3. Experimental results demonstrate the effectiveness and extensive analyses are discussed.

Weaknesses:
1. The experimental results are questioned on fairness, regarding significant tests, varied amounts of training data, adding stronger baselines.
2. The training dataset is solely based on one dataset, which lacks generality.

While there are some issues raised by the reviewers on the experimental design, this paper shows an interesting research idea that might inspire more future studies. Therefore, I lean to the acceptance of this paper, though its score is around the borderline.

**Additional Comments On Reviewer Discussion:**

The discussion focused on several main concerns, including comparison fairness, requests for qualitative examples, and the suggestion to compare against query rewriting approaches. The authors basically handled these concerns during the rebuttal period. Reviewers also actively shared thoughts during the internal discussion period, where one reviewer maintained concerns about comparison fairness. After considering all the feedback, I believe the key technical issues have been clarified and thus recommend the acceptance.

---

### Decision · Program_Chairs · 2025-01-22

Accept (Poster)